# Exploring factors influencing nurses' attitudes towards their role in dental care

**Balgis Gaffar** [1] *, **Eman Bakhurji**[1], **Reem AlKhateeb**[2], **Hussain AlHashim**[2], **Hadeel AlGaoud** [2], **Ziyad AlDaamah** [2], **Jamal AlSaleh** [2], **Rand Aldamanhori**[2], **Shahad AlHamid**[2], **Alanoud AlBarrak**[2], **Intisar Ahmad Siddiqui**[3], **Jorma I. Virtanen** [4,5]

**1** Department of Preventive Dental Sciences, College of Dentistry, Imam Abdulrahman Bin Faisal University, Dammam, Saudi Arabia, **2** College of Dentistry, Imam Abdulrahman Bin Faisal University, Dammam, Saudi Arabia, **3** Department of Dental Education, College of Dentistry, Imam Abdulrahman Bin Faisal University, Dammam, Saudi Arabia, **4** Institute of Dentistry, University of Turku, Turku, Finland, **5** Faculty of Medicine, University of Bergen, Bergen, Norway

* bgosman@iau.edu.sa

## Abstract

### Background

The burden of oral disease requires collaboration between health care professionals. Nurses are frequently exposed to patients and can play a vital role in prevention of oral diseases. This study aimed to investigate nurses' attitudes towards their role in dental care and its associated factors.

### Methods

This cross-sectional, survey-based study recruited a total of 525 nurses in Eastern Saudi Arabia, using a snow-ball sampling technique. Data was collected using an online, pre-validated 40-item questionnaire with four domains (attitudes, knowledge, practices, and demographic data).

### Results

Nurses showed satisfactory attitudes (52.8 ± 8.2) and average knowledge (13.4 ±3.9). More than two thirds (70.3%) reported responding to patients' questions about oral health conditions, 47.1% and 19.7% conducted oral health screening and education respectively. Referral practices were reported by 266 (50.7%) of the participants with pain being the main reason for referrals. Attending lectures/ trainings on oral health and having a formal education about oral health were significantly associated with nurses' positive attitudes towards their role in dental care (P<0.001). Conducting oral health screening or education were also significantly associated with positive attitudes (P = 0.001).

### Conclusion

Positive attitudes were observed among those with undergraduate oral health backgrounds, in continuous education, and those who were involved in oral health screenings.

**Data Availability Statement:** All relevant data are within the paper and its Supporting Information files.

**Funding:** The author(s) received no specific funding for this work.

**Competing interests:** The authors have declared that no competing interests exist.

There is a need for under and postgraduate inclusion of oral health training in nurses' education.

## Introduction

Oral diseases constitute a huge burden on people's daily lives, with dental caries and periodontal diseases being the most prevalent on a global basis [1, 2]. Optimal health cannot be achieved without good oral health with the later impacting individuals' wellbeing and quality of life. In Saudi Arabia, the prevalence and severity of dental caries is high [3], reaching 80 to 90% among children 15 and younger [4] similarly, the prevalence of periodontal disease in Saudi Arabia is higher compared to western countries [5], with recent estimates up to 50% [6]. The burden, distribution, and consequences of oral disease in the community necessitates coordination between all health care providers [7]. Nurses are frequently exposed to patients and are involved in all aspects of healthcare provision from health education, promotion, screening to simple treatments [8] as such they can play a vital role in the prevention and early detection of oral diseases [7, 9–12].

Studies evaluating nurses' knowledge, participation, and attitudes about dental care for patients showed contradictory findings [12–16]. Nurses were reported in some studies to participate in the oral care and prophylaxis of hospitalized and nursing homes patients [12–15]. While Ahmed et al., observed that nurses did not regularly refer expecting mothers or children for dental check-ups and did not inform them about the importance and need of oral hygiene and dental care [17]. Where else in India it was found that nurses despite the lack in basic oral health knowledge, they were still found to have positive attitudes [16].

In the same context, Andargie and Kassahun in a study conducted in Ethiopia found that nurses had inadequate knowledge and attitudes towards dental care with the later being influenced by factors such as level of education, experience, and oral health training [18].

Good oral health knowledge among nurses is essential to provide patients with proper dental care; however sound knowledge may not lead to sound practices unless nurses exhibit positive attitudes towards their role in dental care. A previous study from Eastern Saudi Arabia (KSA) reported that nurses had the most negative attitudes among health professionals [19] and reported e.g. heavy workloads, lack of skills, and knowledge of oral care as barriers that hindered their involvement in dental care [10, 19]. On the other hand, Al-Jobair and colleagues concluded that pediatric nurses in Riyadh had positive attitudes to provide dental care to hospitalized children despite their limited oral health knowledge [20].

The theories of reasoned action and planned behavior have always been referred to in explaining the interplay between knowledge, attitudes, and practices. Initially it was thought that the link between knowledge and behavior/practice is a linear progression, the more the knowledge the more are the favorable behaviors/practices [21]. On the other hand, the theory of planned behavior postulated that it is the person's attitude that will lead to a certain behavior which is usually mediated by previous experience (positive or negative) [22]. From a psychological point of view, attitude is treated as an invisible construct influencing a person's decisions to act in a particular way [23]. As such, it is crucial to explore the factors that influence nurses' attitudes regarding their active role in individuals' oral health. We hypothesized that nurses with sound knowledge or those who were engaged in regular dental care would have more positive attitudes. Therefore, this study aimed to investigate nurses' attitudes towards their role in dental care and factors associated with it.

## Methods

### Study design and setting

This was a cross-sectional, survey-based study conducted in Eastern Saudi Arabia from September to October 2021. The Eastern Province, with more than 36% of Saudi Arabia's total area, is the largest province and the third most populous. The administrative and territorial division of the Eastern Province includes 10 districts: the largest towns include Dammam, Al-Hasa, Al-Jubail, Ras Tanura, Dhahran, Al-Khobar, and Al-Qatif.

### Study participants

The study included registered nurses practicing in Eastern Saudi Arabia in both the public and private sectors, who agreed to participate in the study.

### Sample size and sampling technique

Sample size was calculated to be 377 nurses, based on an assumption of 50 percent of nurses with the correct knowledge and practices to control oral disease, and an estimated number of practicing nurses of about 20,000: a margin of error 5% and confidence level 95% [24].

### Data collection tool

A snowball sampling technique was implemented. We collected data using a pre-validated 40-item questionnaire developed from the literature [11, 12, 14, 16, 25] and focus group discussions. The questionnaire was pilot tested before commencing the study on 15 nurses who were not part of the main study. The questionnaire had four domains: demographic and background information, knowledge, attitudes, and participants' current practices. The questionnaire was distributed online through social media (WhatsApp, Twitter, and Facebook) and was administered in both Arabic and English languages. Given the nature of the sampling technique the distribution of the online survey was done by multiple participants at the same time. The survey link was kept active for one month and all responses were then considered. The study's independent variables were nurses' demographics, oral health knowledge and practices, while the study's outcome was nurses' attitudes about their role/participation in the provision of dental care.

**Demographics and background information.** These included 1) Participants' age, categorized as the twenties (from 20 to 29 years old), the thirties (from 30 to 39 years old), the forties (from 40 to 49 years old), and the fifties and above. 2) Gender was answered as male or female. 3) Current affiliation as to where participants choose public hospitals, the Ministry of Health, teaching institutes, private hospitals, or both the private and public sectors. 4) Years of experience categorized into less than 3 years, between 3 and 6 years, more than 6 years and less than 10 years, or more than 10 years. Participants were also asked about the source of their oral health information by choosing one or more of the following options: formal education (as part of their nursing studies), social media, scientific publications, ministry of health website/materials or can choose no previous oral health knowledge.

**Assessment of knowledge.** Nurses' knowledge about oral health was assessed with 13 questions, and each correct answer was scored as one point, but the wrong answer was scored as zero. The overall knowledge score was determined by the sum of all correct answers based on 25 points. Given the mean, scores were categorized into good knowledge (20–25), average knowledge (12.5–19), and poor knowledge (less than 12.5)

**Assessment of practices.** Nurses' dental care practices were assessed through four open-ended questions. These included: 1) if they provided oral health education; 2) if they provided oral screening questions (answered as always, sometimes, or never); 3) if they responded to

patients' questions about oral health problems as yes or no; 4) if they referred patients to dentists and answered as yes or no. Those who referred patients to dentists were also asked about reasons for the referral.

**Assessment of attitudes.** Nurses' attitudes were assessed through 14 statements using a Likert scale. The Likert score ranged from five points for strongly agreeing to one point for strongly disagreeing, with the total score being 70 points. Based on the mean, attitude scores were categorized into positive (56 or more) or negative (less than 35).

## Statistical analysis

Data analyses were performed using SPSS version 22.0 (IBM Corp., Released 2013 or IBM-SPSS Statistics for Windows, Version 22.0., Armonk, NY, IBM Corp.). We presented descriptive data in the form of frequencies/percentages and/or mean (±SD). The Chi-square test was applied to compare demographic characteristics, knowledge, and oral health practices between groups of participants with positive or negative attitudes towards this realm. A P-value of $\leq 0.05$ was considered statistically significant.

## Results

A total of 525 participants responded to the survey. In our sample most participants 255 (48.6%) were in their twenties, 419 were males (79.8%), and 381 were Saudis (72.6%). The Ministry of Health hired most employees (42.1%) with less than 3 years of experience, while 267 (50.9%) reported having an oral health undergraduate curriculum, and 305 (57.8%) did not attend any educational establishment in oral health.

**Fig 1** shows the sources of oral health information (knowledge) for study participants. Formal education (38%) was followed by social media (18.5%) as the main sources of oral health knowledge, with 11.7% reporting no previous oral health knowledge.

**Table 1** shows participant's attitude towards oral health. The mean (±SD) attitude of participants was 52.8 (± 8.2), indicative of favorable attitudes towards oral health. The most positive attitude regarding the impact of oral health on overall well-being (454, 86.8%). Participants were positively inclined to get trained in oral health education (398, 76.4%) as well as getting

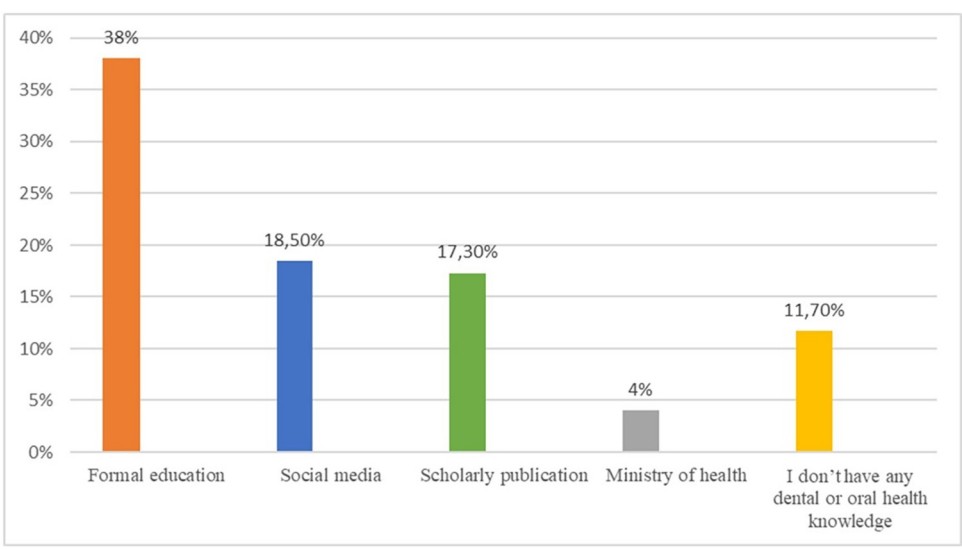

**Fig 1. Sources of oral health information as reported by the study participants.** (N = 525).

**Table 1. Participant's attitude towards oral health.** (N = 525).

| Attitude | Agree (%) | Disagree (%) | Neutral (%) |
|---|---|---|---|
| Oral health affects the overall individual's well-being. | 454 (86.8) | 14 (2.7) | 55 (10.5) |
| Oral health of patients is the duty of all healthcare providers. | 370 (70.5) | 51 (9.7) | 102 (19.4) |
| I have ethical obligations to provide oral health education. | 372 (70.9) | 42 (8.0) | 110 (21.0) |
| It is easy to provide oral health education. | 382 (72.8) | 38 (7.2) | 105 (20.0) |
| I am aware of oral diseases risk factors. | 349 (66.7) | 50 (9.6) | 124 (23.7) |
| It is easy to conduct an oral health screening. | 335 (63.8) | 72 (13.8) | 114 (21.9) |
| I am knowledgeable of how to conduct an oral health screening. | 281 (53.5) | 93 (17.7) | 148 (28.2) |
| I am confident to conduct an oral health screening. | 299 (57.0) | 79 (15.0) | 146 (27.9) |
| I will be overwhelmed if I provided oral health education or screening. | 272 (51.8) | 95 (18.1) | 148 (28.7) |
| I am willing to be trained to provide oral health screening. | 377 (72.0) | 51 (9.8) | 95 (18.2) |
| I am willing to be trained to provide oral health education. | 398 (76.4) | 38 (7.3) | 85 (16.3) |
| I am willing to be trained to provide oral health screening. | 388 (74.3) | 48 (9.2) | 86 (16.5) |
| I am willing to provide oral health education or screening only if I am extra paid. | 250 (47.8) | 135 (25.7) | 138 (26.4) |
| I am willing to provide oral health education or screening only if I have lesser number of patients. | 239 (45.8) | 136 (26.1) | 147 (28.2) |
| I have no time to provide oral health education or screening. | 170 (32.4) | 195 (37.3) | 159 (30.3) |

Values given in parentheses are percentages.

trained to provide oral health screenings (388, 74.3%). Slightly more than half of participants felt knowledgeable or confident enough to provide oral health screening (281, 53.5%) and (299, 57.0%) respectively. An almost equal number of participants agreed they were willing to provide oral health education or screening if they were paid extra (250, 47.8%), with a lower number of patients (239, 45.8%). On the other hand, 195 participants (37.3%) felt there was no time to provide oral health education or screening.

Table 2 indicates that no demographic characteristics were associated with nurses' attitudes on their role in dental care.

The mean (±SD) knowledge score among participants was 13.4 (±3.9), indicative of an average knowledge level on oral health. Table 3 shows the association between participants' oral health knowledge, previous oral health knowledge, training, and their attitudes.

Regarding nurses' knowledge about dental caries, their best scores were about how they could be prevented for 484 participants (92.2%), that caries can occur at any age, and that dental caries may lead to tooth loss, as seen with 445 participants (84.8%). Knowledge of these factors were significantly associated with positive attitudes among nurses ($P \leq 0.05$) (Table 4). In terms of nursing awareness of periodontal disease, the highest knowledge score pertained to periodontitis leading to tooth loss in 421 participants (80.2%), and how gingivitis affects 375 participants (71.4%), the latter showing a statistically significant association with nurses' attitudes ($P = 0.029$) (Table 4). Attending lectures on oral health along with a formal education was associated with nurses' positive attitudes about their role in dental care ($P \leq 0.001$).

Table 4 shows nurses' current practices in terms of dental care and its association with attitudes. More than two thirds of nurses (70.3%) replied that they responded to patients' questions about oral health conditions, but that this was not associated with positive attitudes towards dental care ($P = 0.108$). Yet, conducting oral health screening with an education was practiced to a lesser extent by nurses, at 47.1% and 19.7%, respectively; these practices were significantly associated with positive attitudes among nurses ($P = 0.001$).

Referral practices were reported by half the participants (266, 50.7%); reasons for referrals are shown in Fig 2; however, all referral practices were not necessarily associated with nurses' attitudes (Table 4).

**Table 2. Association of participants' attitude towards oral health with their demographic characteristics.**

| Demographic characteristics | | Total (N = 525) | Positive (N = 220) | Negative (N = 305) | P-value |
|---|---|---|---|---|---|
| Age (years) | 20–29 years old | 255 (48.6) | 97 (44.1) | 158 (51.8) | 0.377 |
| | 30–39 years old | 193 (36.8) | 88 (40.0) | 105 (34.4) | |
| | 40–49 years old | 56 (10.7) | 25 (11.4) | 31 (10.2) | |
| | ≥ 50 years old | 21 (4.0) | 10 (11.4) | 11 (3.6) | |
| Gender | Male | 419 (79.8) | 175 (79.5) | 244 (80.0) | 0.898 |
| | Female | 106 (20.2) | 45 (20.5) | 61 (20.0) | |
| Nationality | Saudi | 381 (72.6) | 166 (75.5) | 215 (70.5) | 0.209 |
| | Non-Saudi | 144 (27.4) | 54 (24.5) | 90 (29.5) | |
| Affiliation | Public hospital | 119 (22.7) | 52 (23.9) | 67 (22.0) | 0.482 |
| | Ministry of health | 175 (33.3) | 74 (33.9) | 101 (33.1) | |
| | Teaching institute | 77 (14.7) | 27 (12.4) | 50 (16.4) | |
| | Private hospital | 120 (22.9) | 48 (22.0) | 72 (23.6) | |
| | Both private & public | 32 (6.1) | 17 (7.8) | 15 (4.9) | |
| Experience | <3 years | 221 (42.1) | 89 (40.5) | 132 (43.3) | 0.875 |
| | 3–6 years | 105 (20.0) | 44 (20.0) | 61 (20.0) | |
| | >6 to10 years | 88 (16.8) | 37 (16.8) | 51 (16.7) | |
| | >10 years | 111 (21.1) | 50 (22.7) | 61 (20.0) | |

Values given in parentheses are percentages. Non-significant at $p \leq 0.05$.

## Discussion

The current study investigated nurses' attitudes towards their role in dental care and factors that influenced positive attitudes. Nurses showed overall positive attitudes towards their role in dental care, and the majority were willing to provide oral health education and/or screening. Having an oral health aspect as part of undergraduate curriculum and attending continuous education in oral health typically led to the acquisition of positive attitudes. In the same

**Table 3. Association between participants current and previous oral health knowledge, or training with their attitudes.**

| Learning | | Attitude towards oral health | | | P-value |
|---|---|---|---|---|---|
| | | Total (N = 525) | Positive (N = 305) | Negative (N = 220) | |
| **Knowledge of oral health** | Knowledge of dental caries | 12 (2.3) | 7 (2.3) | 5 (2.3) | 0.987 |
| | Diet is a risk factor | 181 (34.5) | 114 (37.4) | 67 (30.5) | 0.100 |
| | Poor hygiene | 380 (72.4) | 226 (74.1) | 154 (70.0) | 0.300 |
| | Dental caries can occur at any age. | 462 (88.0) | 283 (92.8)* | 179 (81.4) | **0.001** |
| | Dental caries can be prevented. | 484 (92.2) | 292 (95.7)* | 192 (87.3) | **0.001** |
| | Dental caries can lead to tooth loss. | 445 (84.8) | 268 (87.9)* | 177 (80.5) | **0.020** |
| | Prevention from dental caries. | 18 (24.4) | 77 (25.2) | 51 (23.2) | 0.587 |
| | Knowledge about gingivitis. | 92 (17.5) | 57 (18.7) | 35 (15.9) | 0.409 |
| | Risk factors of gingivitis. | 111 (21.1) | 70 (23.0) | 41 (18.6) | 0.323 |
| | Gingivitis affects adults only. | 375 (71.4) | 229 (75.1)* | 146 (66.4) | **0.029** |
| | Gingivitis can lead to tooth loss. | 153 (29.1) | 96 (31.5) | 57 (25.9) | 0.166 |
| | Periodontitis is the same as gingivitis. | 309 (58.9) | 182 (59.7) | 127 (57.7) | 0.655 |
| | Knowledge about periodontitis. | 16 (3.0) | 6 (2.0) | 10 (4.5) | 0.090 |
| | Periodontitis can lead to tooth loss. | 421 (80.2) | 248 (81.3) | 173 (78.6) | 0.448 |
| **Attended lecture/ training on oral health** | | 220 (42.2) | 68 (31.1)* | 152 (50.3) | **0.001** |
| **Have formal education of OH** | | 258 (49.1) | 89 (40.5) | 169 (55.4) | **<0.001** |

**Table 4. Association between participants' dental care practices and their attitudes.**

| Practice of oral health | | Attitude towards oral health | | P-value |
|---|---|---|---|---|
| | Total (n = 525) | Positive (n = 305) | Negative (n = 220) | |
| Always conduct an oral health screening | 103 (19.7) | 69 (22.8)* | 34 (15.5) | **0.001** |
| Sometimes conduct an oral health screening. | 246 (47.1) | 156 (51.5)* | 90 (41.1) | **0.001** |
| Responded to a patient question about an oral health condition. | 367 (70.3) | 222 (73.0) | 145 (73.0) | 0.108 |
| Referred a patient to a dentist. | 266 (50.7) | 163 (53.4) | 103 (46.8) | 0.134 |
| Reason of referral was dental pain. | 194 (37.0) | 118 (38.7) | 76 (34.5) | 0.332 |
| Reason of referral was cavitation. | 87 (16.6) | 54 (17.7) | 33 (15.0) | 0.411 |
| Reason of referral was swelling. | 102 (19.4) | 57 (18.7) | 45 (20.5) | 0.614 |
| Reason of referral was broken tooth. | 91 (17.3) | 56 (18.4) | 35 (15.9) | 0.464 |
| Reason of referral was abscess | 79 (15.0) | 41 (13.4) | 38 (17.3) | 0.226 |
| Reason of referral was facial trauma. | 40 (7.6) | 25 (8.2) | 15 (6.8) | 0.557 |
| Reason of referral was prophylaxis. | 39 (7.4) | 24 (7.9) | 15 (6.8) | 0.651 |
| Did not refer patients before. | 230 (43.8) | 136 (44.6) | 94 (42.7) | 0.671 |

context, nurses who provided oral health screening showed more positive attitudes than those who did not. None of the demographic factors had an influence on attitudes. As such, the study's hypothesis was partially supported. Nurses play an essential role in health care and are the staff with the most exposure to patients; their role in oral health care and promotion cannot be overlooked. Yet, active involvement requires favorable attitudes by promoting this role.

Nurses in the current study had a generally positive attitude towards their role in dental care, with the most favorable attitude on the impact of oral health on individuals' general health and well-being (86.8%), consistent with reports from Finland (100% of all interviewed nurses) [10], Korea (83.2%) [25] and Eritrea (89%) [26]. Poor oral health can aggravate many systemic conditions as well as impairment of children's growth [27] and quality of life for the elderly [28]; this cites the impact of poor oral health on individuals' general health, which can improve the medical care provided and treatment outcomes. Providing proper health care is

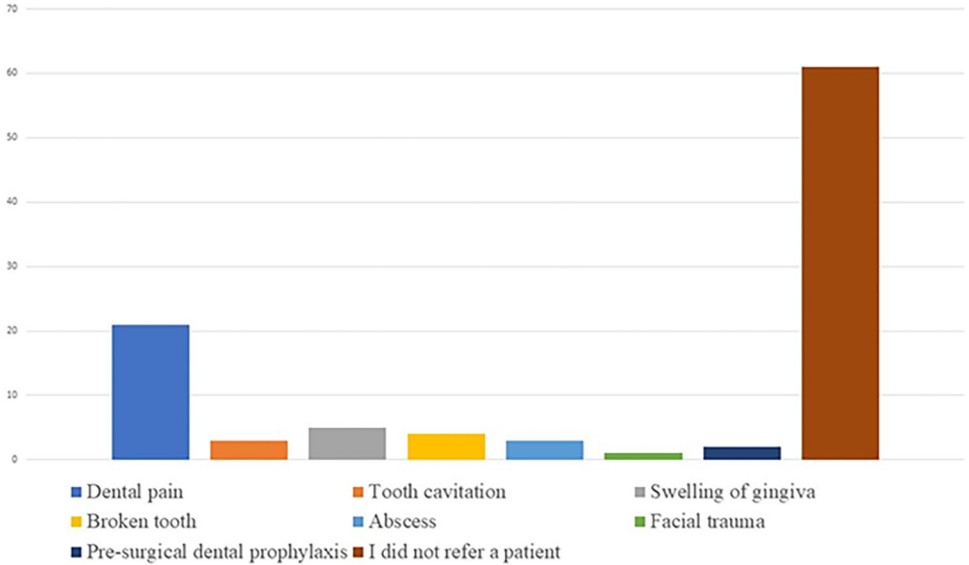

**Fig 2. Reasons for dental referrals as reported by nurses.** (N = 525).

imperative in every medical field, as such efficient collaboration between those in medical practice is fundamental for proper patient services.

Nurses in the current study were willing to provide oral health education and screening, although they had concerns related to workload and financial compensation. Factors such as limited time, number of nurses, and resources have been highlighted as barriers in previous studies [10, 12]. Attitudes can be formed through conditioning, in which consumption of the attitude object is reinforced. One type of reinforcement is called positive reinforcement, in which the response is strengthened, and appropriate behavior occurs when provided with a reward [21].

Demographic factors in the current study showed no association with nurses' attitudes; a similar finding was also reported by Dagnew et al. [26]. The link to demographic or personal characteristics with nurses' attitudes varied in other studies: one conducted in the capital city of Saudi Arabia found that gender, nationality, and previous training were powerful predictors of favorable attitudes towards patient oral health assessment [29]. Another study conducted in Taif (a city in Western Saudi Arabia), found significant gender differences in knowledge and practices, but not attitudes. The same study also found that experience significantly influenced nurses' attitudes [30]. Differences in these reports could be due to varied settings in which the studies were conducted; for example, nurses caring for elderly or psychotic patients may have had more tasks than those in other departments. In terms of attitudes and demographics, younger males and those with less education were usually viewed as risk takers who still showed favorable attitudes not accepted by the majority [31]. Observed differences in the demographic influence on attitudes call for assessments of the environment and organizational support.

Nurses in the current study showed average oral health knowledge. Findings from previous studies have been controversial, whereas some studies observed a lack of oral health knowledge [10, 16]; Shimpi et al. reported a good level of oral health knowledge in nurses [32]. Any short-fall in knowledge could be attributed to a lack of continued oral health training in the work-force, reflected in the lack of confidence to conduct oral health screenings as reported by study participants. Proper oral health knowledge among nurses and awareness of dental disease had a significant implication for patient care, as oral problems could be detected at earlier stages with better prognoses [33]. We found that proper knowledge about dental disease (their clinical manifestations and consequences) was associated with positive attitudes among nurses.

Attending educational sessions on oral health in undergraduate studies was associated with positive attitudes. Philip et al. observed that nurses who received oral health education in university but not in clinical practice had overall less oral health knowledge [16]. We highlight the need to reinforce this knowledge in foundational training and being informed about current best practices, with the need to improve the dental component -if any- of the nursing curriculum.

In our study, half the participants mentioned how they sometimes provided oral health education to patients and referred their patients to a dentist; a majority of participants reported responding to patients' questions about oral health conditions. In a similar study in the USA, more than two thirds claimed they frequently performed oral health assessments [32], while in Saudi Arabia half the nurses assessed patient oral health in their departments [29] while in a Nigerian study, almost all the nurses never referred a patient to a dentist [33]. Sound knowledge is the first step in establishing regular practices [34], which explains the gap in the current study.

Studies conducted in Saudi Arabia highlighted the need for mandatory workshops and informative seminars by the Saudi Ministry of Health, which encouraged nurses to perform oral health screening as part of their daily tasks (given the high prevalence of dental disease in

the country). A previous study found that providing nurses with education on ways to implement oral health assessments increased nursing practice to document oral health assessments of elderly residents [35]. Another factor found to enhance nurses' oral health assessment and care was policy within their institution [29], which can be accurate about patient referrals. In the current study, more than half the nurses did not refer patients to a dentist. The lack of a clear loop for referrals and guidelines was documented as the main barrier that hindered referrals in previous studies [26, 29]. Another factor linked to less oral health assessment and care was inadequate training [30]. This is seen at both undergraduate and graduate levels, while onsite and online workshops provided nurses with skills to guide their practices.

There are a few limitations in the current study that we want to acknowledge. The cross-sectional design of the study only corroborates an association, but not causation. However, this design is most appropriate to investigate current attitudes, knowledge, and behavior. This study relied on self-reported data, which can lead to overreporting of favorable practices. Further studies may use patient's medical records to record these practices to confirm report authenticity. Also, differences in quantifying and categorizing knowledge between studies may contribute to conflicting results. Although the nature of sampling techniques may result in some self-selection bias, this design was most appropriate to collect large data from different institutions, within a limited time frame. Despite these limitations, the validated questionnaire and the adequate sample size allowed for the generalizability of the study findings. This directs attention to the nurses' role in oral health care in Eastern Saudi Arabia, as well as in decision and policy for future planning.

## Conclusion

Nurses in the current study had positive attitudes, average knowledge, and minimum practices in regards to dental care. Positive attitudes were noted among those who had an undergraduate oral health component, were involved in continuous education, and provided oral health screening. There is a need for inclusion of oral health training in nurses' under- and postgraduate education. Future studies should explore the effects of institutional guidelines/policies and type of department on nurses' attitudes. The integration of oral and general health should be the cornerstone of policy approaches for the prevention and control of oral disease.

## Supporting information

**S1 Data.**
(XLS)

## Acknowledgments

### Ethics approval and consent to participate

This study was approved by the Deanship of Scientific Research–Imam Abdulrahman bin Faisal University (IRB- 201702- 048). An introductory section preceded the questionnaire and debriefed participants about the study's purpose, data collection mode, ensuring participants of the confidentiality of their responses. We confirm that all methods were carried out in accordance with relevant guidelines and regulations and informed consent was obtained from all subjects by ticking a box "I have read the introductory section and agree to participate in the study" and proceed to the survey. Proceeding to the survey after reading the introductory section and clicking "agree" button was considered as a consent to participate. Due to the nature of the online data collection, electronic informed consent was seen as the most feasible

method. The details of consent procedure were described in the study protocol which was reviewed and approved by the ethical committee.

## Author Contributions

**Conceptualization:** Balgis Gaffar.

**Data curation:** Balgis Gaffar, Eman Bakhurji, Intisar Ahmad Siddiqui.

**Formal analysis:** Eman Bakhurji, Intisar Ahmad Siddiqui.

**Investigation:** Reem AlKhateeb, Hussain AlHashim, Hadeel AlGaoud, Ziyad AlDaamah, Jamal AlSaleh, Rand Aldamanhori, Shahad AlHamid, Alanoud AlBarrak.

**Methodology:** Reem AlKhateeb, Hussain AlHashim, Hadeel AlGaoud, Ziyad AlDaamah, Jamal AlSaleh, Rand Aldamanhori, Shahad AlHamid, Alanoud AlBarrak, Jorma I. Virtanen.

**Writing – original draft:** Balgis Gaffar, Reem AlKhateeb, Hussain AlHashim, Hadeel AlGaoud, Ziyad AlDaamah, Jamal AlSaleh, Rand Aldamanhori, Shahad AlHamid, Alanoud AlBarrak.

**Writing – review & editing:** Eman Bakhurji, Intisar Ahmad Siddiqui, Jorma I. Virtanen.

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
