## [Decision Letter · Decision Letter 0]

7 Mar 2023

PONE-D-22-32331Factors associated with nurses’ attitudes towards their role in dental care.PLOS ONE

Dear Dr. Gaffar ,

Thank you for submitting your manuscript to PLOS ONE. After careful consideration, we feel that it has merit but does not fully meet PLOS ONE’s publication criteria as it currently stands. Therefore, we invite you to submit a revised version of the manuscript that addresses the points raised during the review process.

We look forward to receiving your revised manuscript.

Kind regards,

Samira Adnan

Academic Editor

PLOS ONE

Journal Requirements:

3. Please amend your current ethics statement to address the following concerns:

a) Did participants provide their written or verbal informed consent to participate in this study?

b) If consent was verbal, please explain i) why written consent was not obtained, ii) how you documented participant consent, and iii) whether the ethics committees/IRB approved this consent procedure

Reviewers' comments:

Reviewer's Responses to Questions

**Comments to the Author**

1. Is the manuscript technically sound, and do the data support the conclusions?

Reviewer #1: Yes

Reviewer #2: Partly

2. Has the statistical analysis been performed appropriately and rigorously? 

Reviewer #1: No

Reviewer #2: No

3. Have the authors made all data underlying the findings in their manuscript fully available?

Reviewer #1: Yes

Reviewer #2: Yes

4. Is the manuscript presented in an intelligible fashion and written in standard English?

Reviewer #1: No

Reviewer #2: Yes

5. Review Comments to the Author

Reviewer #1: The topic is interesting and probably will show good findings if it was designed with an adequate method. The following comments are reported in the review of this article.

Page 3, background, first paragraph: I expect to find some text discussing the objective of the study which is the attitude of the nurses BUT not related to dental caries.

This should be totally changed to cope with the aim of the study rather than the nature of dental caries.

Page 4, Background, line 90: This sentence is not of clear expression.

Page 4, Background lines 91-92: these sentences did not show any correlation with the previous text.

Page 4, Background line 106: The aim of the study was different from what was stated in the abstract, therefore, should be corrected and unified. The one in the abstract probably sounds better.

Generally saying: the introduction was not properly written and shows incoherence between sentences and ideas.

Page 5, Method, sample size calculation, line 127: if the Sample size was calculated to be 377 nurses, based on an assumption of 50 percent of nurses, So what methods were used to inflate the sample to 525 as mentioned in the abstract?

This should be clearly described.

Page 5, Methods, data collection tool, line 135:

How the investigators assured that the respondents were nurses? and how was the process conducted to reach a nurse and not another person?

Page 5, methods, line 138: “while the study’s outcome was based on their attitudes.”

this outcome is vague. When do we say attitude what? should be rephrased clearly.

Page 6, Methods, Assessment of attitude, line 157: The calculation was incorrect based on what was stated in the description. If using 5 scales Likert score for 14 questions with a 0 score for strongly disagree, the numbers will be 0, 1, 2, 3, and 4, and no number 5. So, the total score will be 56 and not 70 as mentioned in their description. This will totally change the calculation of the attitude level and basically will move all the results that need to be re-calculated in the study.

Page 7, Results, table 1: How the investigator approached those non-Saudi nurses, as they used the same version of the questionnaire or a different language?

Page 8, line 181:

Results, Figure 1 is demonstrating the source of oral health information which was not mentioned in the methods. and figure 2, the data was not clear.

Page 9, results, line 202: The total inserted in table 3 is the repetition of the same result in table 1. Therefore, it is better to stick to one of the findings and omit the other.

Page 10, Results, line 221: based on the wrong calculation of the total score and the categorical score of the attitude, the rest of the findings will be inadequate. Therefore, it needs to repeat the cut-off of goodness for the attitude and then again make your analysis on the attitude towards oral health section.

Discussion, page 13, line 241: "Saudi nurses showed overall positive attitudes towards their role in dental care, and the majority were willing to provide oral health education"

This fact should be revised after the correction of the calculation of the cut-off point and the scores used in this part.

Discussion, page 13, line 255: as the author stated that the positive role of the nurses is comparable with other studies is good but also needs to show the consistency or differences with other studies by percentage at least for each study.

Limitation, page 16: among the limitations that were found in the study is the use of online distribution of the questionnaire for a specific profession without control whether they are students in the dental college or other general population or even from outside the country. This part showed an inflated sample size without any clarification of the reason except it is a snowball method.

References, 9, and 10 no page numbers while reference #24 need revision for the year.

Reviewer #2: Relevant and interesting option to improve the oral health of community. There are few suggestions and 1 necessary change.

1. The introduction of the abstract require some connection between 1st and 2 nd sentence. Both sentences are misfit when read in continuity.

2. Is the questionnaire self designed? if yes then did the authors conduct pilot study to validate it?

3. Title also requires change.

6. PLOS authors have the option to publish the peer review history of their article (what does this mean?). If published, this will include your full peer review and any attached files.

Reviewer #1: **Yes: **Amen A Bawazir

Reviewer #2: No

---

## [Author Response · Author response to Decision Letter 0]

15 May 2023

Reviewer 1

We thank the Reviewer for the positive feedback and valuable comments.

We thank the Reviewer for the valuable suggestion. We have amended the introduction and elaborated more on nurses’ attitudes.

We have now amended the sentence to clarify it.

We have amended the sentences. 

We thank the Reviewer for the comment. We have now corrected and unified the aim in line with the abstract.

We have now amended the introduction completely as suggested by the Reviewer.

We used a snowball sampling technique; the distribution/sharing of the online survey was done spontaneously by multiple respondents at the same time. As such we did not have estimates on responses as for those of face to face or one to one recruitment. We kept the survey link active for one month and all responses were then considered.

We added this clarification to the method section (Page 6 lines 137-140). 

Through a short-listed personal connection (collected from hospital websites) we reached out for nurses who were requested to share the survey link with registered nurses practicing in Eastern Saudi Arabia only. Any responses from other health professionals were excluded.

We thank the Reviewer for catching this, the sentence was incomplete and is now rephrased (Page 6 lines 141&142).

We thank the Reviewer for catching this. We did use a 5-point Likert scale with score of five for strongly agreeing and score one for strongly disagreeing; there was a mistake in the previous method part (a score from 0 to 4). All results presented are based on the scores from 1 to 5.

We have now corrected the method section (Page 7 line 170).

The survey was administered in both Arabic and English languages. Each question was written in Arabic and English. 

We added this to the methodology (Page 6 lines 137-139). 

We thank the Reviewer for catching this. We added the sources of oral health knowledge (figure 1) as a part of background information (Page 6 lines 151-154). 

Figure 2 is now presented as a bar chart.

Table 1 has now been removed to avoid repetition.

As previously explained this was a typographic error. The results are based on the correct scoring from 1 to 5.

The cut-off point is correct, as mentioned in response to the Reviewer’s previous comment. The typographic mistake has now been corrected (Page 7 line 170).

Percentages were added for the studies from Finland and Ethiopia. Since the study from Australia did not report actual percentage it was replaced by a study from Korea (Lee YJ, et al.(2019). 

We thank the Reviewer for the comment. We have clarified in the method section according to your previous comments that the study included only nurses. The inflated sample was mainly due to the nature of the sampling technique.

We added the page numbers to references 9 (now reference #6) and 10 (now reference #7).

The year was corrected for reference #24 (now reference # 27).

Reviewer 2

We thank the Reviewer for the positive feedback.

We have now amended the introduction as suggested by the reviewer. 

We conducted a pilot study, and this has been added to the method section (Page 6 lines 133-135).

The title was modified to “Exploring factors influencing nurses’ attitudes towards their role in dental care”

---

## [Decision Letter · Decision Letter 1]

7 Jul 2023

Exploring factors influencing nurses’ attitudes towards their role in dental care

PONE-D-22-32331R1

Dear Dr. Gaffar ,

We’re pleased to inform you that your manuscript has been judged scientifically suitable for publication and will be formally accepted for publication once it meets all outstanding technical requirements.

Kind regards,

Samira Adnan

Academic Editor

PLOS ONE

Additional Editor Comments (optional):

Reviewers' comments:

Reviewer's Responses to Questions

**Comments to the Author**

1. If the authors have adequately addressed your comments raised in a previous round of review and you feel that this manuscript is now acceptable for publication, you may indicate that here to bypass the “Comments to the Author” section, enter your conflict of interest statement in the “Confidential to Editor” section, and submit your "Accept" recommendation.

Reviewer #1: All comments have been addressed

Reviewer #2: All comments have been addressed

2. Is the manuscript technically sound, and do the data support the conclusions?

Reviewer #1: Yes

Reviewer #2: Yes

3. Has the statistical analysis been performed appropriately and rigorously? 

Reviewer #1: Yes

Reviewer #2: Yes

4. Have the authors made all data underlying the findings in their manuscript fully available?

Reviewer #1: Yes

Reviewer #2: Yes

5. Is the manuscript presented in an intelligible fashion and written in standard English?

Reviewer #1: Yes

Reviewer #2: Yes

6. Review Comments to the Author

Reviewer #1: The correction and changes made was adequate and going with the raised comments.

In addition, it was made according to the standard of the Journal

Reviewer #2: Thank you so much for incoporating the suggestions.The overall quality of the manuscript has improved.

7. PLOS authors have the option to publish the peer review history of their article (what does this mean?). If published, this will include your full peer review and any attached files.

Reviewer #1: **Yes: **Amen A. Bawazir

Reviewer #2: No

---

## [Editor Report · Acceptance letter]

11 Jul 2023

PONE-D-22-32331R1 

Exploring factors influencing nurses’ attitudes towards their role in dental care. 

Dear Dr. Gaffar :

I'm pleased to inform you that your manuscript has been deemed suitable for publication in PLOS ONE. Congratulations! Your manuscript is now with our production department. 

Kind regards, 

on behalf of

Dr. Samira Adnan 

Academic Editor

PLOS ONE